# Towards bioresource-based aggregation-induced emission luminogens from lignin β-O-4 motifs as renewable resources

Tenglong Guo[1,5], Yuting Lin[2,5], Deng Pan[3,5], Xuedan Zhang[2], Wenqing Zhu[1], Xu-Min Cai [2]✉, Genping Huang [3]✉, Hua Wang[1], Dezhu Xu[1], Fritz E. Kühn [4], Bo Zhang [1]✉ & Tao Zhang [1]✉

One-pot synthesis of heterocyclic aromatics with good optical properties from phenolic β-O-4 lignin segments is of high importance to meet high value added biorefinery demands. However, executing this process remains a huge challenge due to the incompatible reaction conditions of the depolymerization of lignin β-O-4 segments containing γ-OH functionalities and bioresource-based aggregation-induced emission luminogens (BioAIEgens) formation with the desired properties. In this work, benzannulation reactions starting from lignin β-O-4 moieties with 3-alkenylated indoles catalyzed by vanadium-based complexes have been successfully developed, affording a wide range of functionalized carbazoles with up to 92% yield. Experiments and density functional theory calculations suggest that the reaction pathway involves the selective cleavage of double C-O bonds/Diels-Alder cycloaddition/dehydrogenative aromatization. Photophysical investigations show that these carbazole products represent a class of BioAIEgens with twisted intramolecular charge transfer. Distinctions of emission behavior were revealed based on unique acceptor-donor-acceptor-type molecular conformations as well as molecular packings. This work features lignin β-O-4 motifs with γ-OH functionalities as renewable substrates, without the need to apply external oxidant/reductant systems. Here, we show a concise and sustainable route to functional carbazoles with AIE properties, building a bridge between lignin and BioAIE materials.

Both the damaging environmental impact and the increasing depletion of fossil resources have triggered strong interest in utilizing biomass for helping to satisfy the growing energy demand in a sustainable form as well as a source for basic chemicals[1,2]. As a high volume and fairly cheap renewable resource, lignin is a unique precursor for the production of aromatic chemicals[3,4]. With respect to structural moieties, β-O-4 (β-aryl ether) associated with γ-OH is the most abundant linkage in lignin, making up approximately 50–65% of all linkages, dependent

[1]CAS Key Laboratory of Science and Technology on Applied Catalysis, Dalian Institute of Chemical Physics, Chinese Academy of Sciences, Dalian 116023, China. [2]Jiangsu Co-Innovation Center of Efficient Processing and Utilization of Forest Resources, International Innovation Center for Forest Chemicals and Materials, College of Chemical Engineering, Nanjing Forestry University, Nanjing 210037, China. [3]Department of Chemistry, School of Science and Tianjin Key Laboratory of Molecular Optoelectronic Sciences, Tianjin University, Tianjin 300072, China. [4]Molecular Catalysis, Catalysis Research Center and Department of Chemistry, Technical University of Munich, Lichtenbergstr. 4, D-85748 Garching bei München, Germany. [5]These authors contributed equally: Tenglong Guo, Yuting Lin, Deng Pan. ✉e-mail: xumin.cai@njfu.edu.cn; gphuang@tju.edu.cn; bo.zhang@dicp.ac.cn; taozhang@dicp.ac.cn

**Fig. 1 | N-participated depolymerization of lignin phenolic β-O-4 motifs containing γ-OH groups. a** Structures of lignin β-O-4 motifs; **b** synthesis of N-containing aromatics from lignin β-O-4 with γ-OH through two steps; **c** one-pot synthesis of N-containing aromatics from lignin β-O-4 with γ-OH; **d** construction of carbazole-based BioAIEgen from lignin β-O-4 with γ-OH.

on the lignin type (Fig. 1a)[5]. Effective depolymerization of this particular linkage could not only remove a considerable barrier in the degradation of lignin feedstocks, but also provide useful depolymerized molecules as scaffolds for potential applications. Thus, extensive efforts have been dedicated to deconstruct the β-O-4 motif to deliver C, H, O-containing products with additional hydrogen and oxygen donating agents. With the aim to functionalize lignin downstream products further, introduction of nitrogen to form nitrogen-containing aromatics is also emerging and increasingly brought into focus[6–8]. So far, most of the current strategies are limited to the

employment of lignin-derived monophenols or modified β-O-4 dimers as starting materials to produce N-containing aromatics in the presence of a N-source through one or multiple steps[9–19] (Fig. 1b, Routes 1–2). Additionally, hydrogenation and oxidation processes are unavoidable. As a challenge, the direct transformation of phenolic β-O-4 linkages to N-containing aromatics without external H or O source is still prominent. Building on recent progress, our group and that of Jiao independently developed methods for the preparation of various N-containing aromatics (benzylamines, pyrimidines, quinolines, quinoxalines and anilines) with organic amines or NaN₃ as N-sources through controlled cleavage of the C-O bond in β-O-4 motifs and the construction of C-C/C-N bonds in a one-pot fashion (Fig. 1c, Route 3)[20–24]. However, these systems exhibit low efficiency and poor selectivity when meeting β-O-4 moieties containing γ-OH groups, leading to low yields of N-containing aromatics (<40%). The higher bond dissociation energies of β-O-4 linkage with γ-OH (69.2 kcal/mol)[25] compared with β-O-4 without γ-OH (65.5 kcal/mol)[26] leads to a variety of unwanted by-products, which is the main reason of lower yield. Moreover, the target molecule application is so far largely confined to the synthesis of medicinal intermediates, large volume application is prohibited by the need of further functionalization. Therefore, the development of a simple and efficient strategy for the production of N-containing aromatics would lead to new applications of lignin-based chemicals.

Bio-resource derived aggregation-induced emission luminogens (BioAIEgens, AIEgens or AIE-active nano agents obtained from natural resources including natural products or derivatives by modifying natural products)[27–30] have raised considerable attention owing to their advantageous renewability, biodegradability, and biocompatibility, paving the way to potential applications in biomedical[31–33], chemosensors/biosensors[34,35], and optoelectronic devices[36,37]. As a natural biomass rich in aromatic and carbonyl subunits, lignin based building blocks display strong fluorescence and excellent self-assembly properties, which could make them serve as ideal precursors for the preparation of BioAIEgens[33,38–40]. However, its complex structure makes it difficult to controllably degrade, thereby limiting the application of lignin. Current studies are usually limited to macromolecular BioAIEgens directly coming from lignin itself or structurally modified lignin based macromolecules, resulting in the fact that the luminescence mechanism and structure-property relationships are still unclear due to the complicated and uncontrollable macromolecular structures. Therefore, the efficient and controllable conversion of lignin platform chemicals into value-added fine chemicals has become a hot research topic for utilization of lignin. Therefore, obtaining BioAIEgens with well-defined structures through lignin depolymerization would provide an alternative strategy for the design of lignin-derived BioAIEgens. Conventional state of the art design of AIEgens is based on restriction of intramolecular motion[41]. Moreover, the construction of electronic donor-acceptor (D-A) configuration is regarded as an effective approach to manipulate the properties of luminescent materials[42,43]. One type of D-A structured molecules, namely the D-A-D type, has attracted considerable interest[44–46]. In contrast, reports on A-D-A molecules as luminescent materials are relatively limited. In recent years, some reports suggest that A-D-A compounds can exhibit stronger charge transfer effects and light-harvesting capability compared to D-A-D structure, hence holding great potential in organic photovoltaics and biomedical applications[47–50]. BioAIEgen associated with A-D-A configuration may synchronously regulate the photophysical properties. Hence, developing a concise and sustainable route to construct BioAIEgens with A-D-A configuration would be a great achievement and highly desirable as potential phototheranostic agents.

Carbazole is an important N-containing heterocyclic organic compound with low ionization potential and a strong electron donor ability[51], these features make them as an ideal electron donor to construct D-A structures[52–55]. Employing carbazole as a core moiety, introducing two electron acceptor groups (i.e., benzoyl) into a carbazole scaffold might provide an efficient way for the preparation of A-D-A-type AIEgens. Moreover, extensive synthetic strategies of the functionalized carbazoles focus on intermolecular cross-coupling reactions between C-H/C-X bonds (X = halo, N, O, C, etc)[56], oxidative intramolecular C-H/C-H cross coupling of prefunctionalized diarylamines[57], the construction of a benzene ring upon substituted indoles through transition metal catalysis[58] or Brønsted acid catalysis[59–61]. However, most approaches suffer from the need of excess oxidants, multi-step reactions and the formation of non-renewable substrates, which raises atom economy and environmental concerns. Thus, a sustainable protocol for producing carbazoles synthesis is desirable.

In this work, we developed a sustainable route for the production of functionalized carbazoles through transformation of lignin β-O-4 segments. γ-OH groups of β-O-4 could be well tolerated. Furthermore, the establishment of the triangular A-D-A skeleton, based on the electron donor ability of carbazole and the corresponding electron acceptor behavior of benzoyl resulting from this unique synthetic methodology has successfully led to the desired twisted intramolecular charge transfer (TICT) AIE properties (Fig. 1d, Route 4).

## Results and discussion
### Reaction optimization
Initially, β-O-4 linkages containing a γ-OH group (1a), highly abundant in natural lignin, are selected as a probe reaction with alkenylated indole 2a catalyzed by vanadium associated with tridentate Schiff base ligands (V-based complexes)[62]. The results show that carbazole derivative 3a as a target product forms in 89% yield along with guaiacol (4a) in 99% yield (Supplementary Table 1, entry 1), demonstrating that carbazole formation occurs, associated with a selective C-O bond cleavage of the lignin model compound, C-C bond construction and dehydrogenation/aromatization in a one-pot fashion. Based on this exciting reactivity, the experiment parameters (catalyst loading, substrates ratio, and reaction temperature) are optimized (Supplementary Tables 1–3). Screening various solvents reveal that the reaction proceeds most efficiently in toluene, compared to N, N-dimethylformamide (DMF), CH₃OH, CH₃CN, tetrahydrofuran and H₂O (Supplementary Table 1, entries 1–6). Additionally, the reaction is very sensitive to reaction temperature. Upon lowering the temperature, yield of 3a decreases significantly from 89% at 140 ˚C to 76% at 120 °C (Supplementary Table 1, entries 1, 7–8). In contrast, changing the catalyst to commercial VO(OEt)₃ and 2,2′-bipyridine ligand, the carbazole 3a is formed only in 55% yield (Supplementary Table 1, entry 9). Notable, a blank experiment shows that without catalyst no product is formed (Supplementary Table 1, entry 10), illustrating the V-based catalyst indeed plays a crucial role.

### Substrate scope
A broad spectrum of substrates with varying electronic and steric effects, and various functional groups on the aromatic ring of the lignin model compound 1 and alkenylated indoles 2 in the presence of V-based catalyst is explored, affording good to excellent yields of the corresponding carbazole derivatives 3. As shown in Fig. 2, lignin β-O-4 model substrates 1a and 1b bearing methoxy substituents on the aryl moieties associated with 2a as a reaction partner to generate the target carbazole products 3a and 3b lead to 91% and 86% isolated yield, along with 99% and 85% yield of guaiacol 4a, respectively (Fig. 2, entries 1–2). To our delight, the reaction also tolerates a hydroxyl group on the aryl ring of 1c, yielding 71% of 3c (Fig. 2, entry 3). Compound 1d bearing no functional group leads to a somewhat reduced yield of carbazole 3d (61%) and of phenol 4b (76%) (Fig. 2, entry 4), illustrating that methoxyl groups on the aryl ring of 1 execute a positive influence on the reaction efficiency. It is noteworthy that N-induced deconstruction of lignin β-

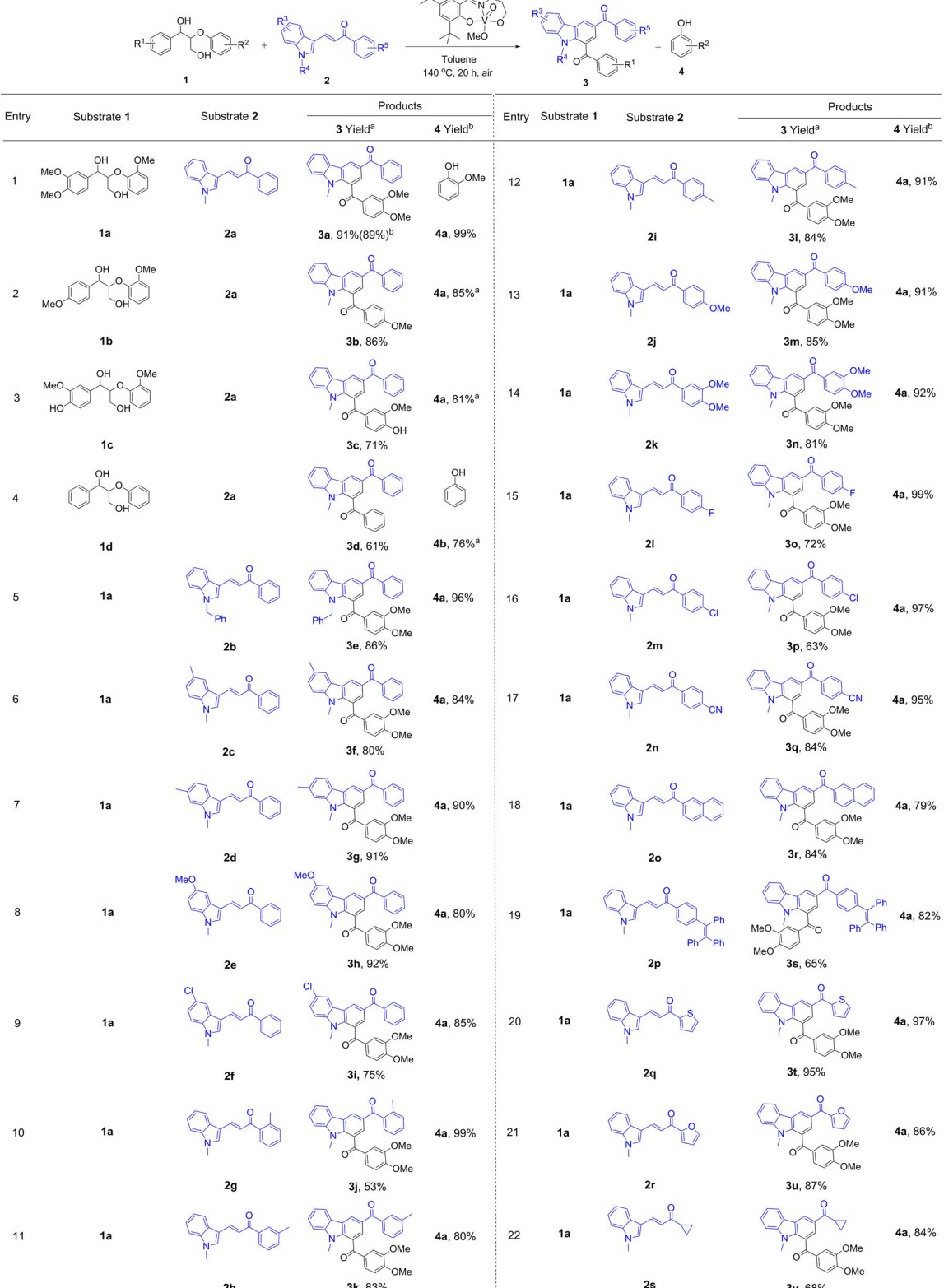

**Fig. 2 | Synthesis of carbazole derivatives** 3 **with various substrates** 1 **and** 2. Reaction conditions: **1** (0.4 mmol), **2** (0.2 mmol), V-complex catalyst (10 mol%), toluene (4 mL) in air, 140 ˚C, reaction time (t) = 20 h, yields of **3** and **4** are calculated based on the amounts of **2** and **1**, respectively. [a]Isolated molar yield. [b]Yields are determined by HPLC with an external standard method.

O-4 linkages containing γ-OH group generally produces lower yields (<40%) of N-containing aromatics due to the complicated structure and higher steric hindrance[20–24]. With the catalytic system applied here, β-O-4 motifs with γ-OH **1a**–**1d** are efficiently transformed to carbazole derivatives **3a**–**3d** in high isolated yields (61–91%). In contrast, the reaction of a N-H free indole derivative ((E)-3-(1H-indol-3-yl)-1-phenyl-prop-2-en-1-one) with **1d** is carried out, leading to ((E)-3-(1-(3-oxo-3-phenylpropyl)-1H-indol-3-yl)-1-phenylprop-2-en-1-one as main product

**Fig. 3 | Mechanistic experiments.** (1) selective C-O bond cleavage of **1d** catalyzed by V-complex catalyst; (2)–(3) the reactions with and without V-complex catalyst under air; (4)–(5) the reactions with and without V-complex catalyst under argon environment.

in 21% yield due to some side reactions (Supplementary Fig. 1). In order to obtain high selectivity of carbazole products, N-protected indole derivatives (**2a**–**2s**) are used as substrates. For example, a benzyl substituent on the N-R moiety in 3-alkenylated is tolerated, leading to 86% yield of **3e** (Fig. 2, entry 5). Additionally, electron-donating substituents such as –Me and -OCH$_3$ on the aryl rings of the indole moieties apparently do not affect the reaction efficiency, since **3f**–**3h** are obtained in 80–92% yield (Fig. 2, entries 6–8), whereas an electron-withdrawing Cl- group decreases the yield of product **3i** to 75% (Fig. 2, entry 9). To further investigate the scope of this reaction, various substituents on the phenyl moiety of the alkenylated indoles **2** were applied. The substrates bearing-Me groups in meta- or para- position lead to **3k** and **3l** (83% and 84%) in higher yields compared with a Me group in ortho-position (**3j**, 53%) due to the steric hindrance in the latter case (Fig. 2, entries 10–12). No obvious electronic effect was detected for the substituted group on the aryl moiety of **2** in this transformation. *P*-Me, *p*-OCH$_3$, *p*-F, *p*-Cl, *p*-CN and aryl substituted alkenylated indoles **2** proceed to deliver the expected products **3l**–**3r** in 63–85% yield (Fig. 2, entries 11–18). Moreover, a tribenzene-alkene bearing substrate also react smoothly, giving the target product **3s** in moderate yield (65%) (Fig. 2, entry 19). Notably, heterocyclic functionalities such as thiophene and furan groups also allow the formation of carbazole derivatives **3t** and **3u** in 95% and 87% yield (Fig. 2, entries 20–21). Beyond that, this process is also effective for a substrate

containing an alkyl ring, obtaining target product **3v** in 68% yield (Fig. 2, entry 22). Significantly, the phenol products **4a**–**4b**, being valuable precursors of detergents, pesticides, antimicrobials and pharmaceutical drugs[63], can be obtained in yields of 76–99%. All these results demonstrate that the system described in this work displays excellent tolerance to a broad substrate scope, not only providing access to carbazole derivatives in high isolated yields, but also achieving phenol derivatives in excellent yields, greatly increasing atom-economy.

## Mechanistic studies

With the aim to elucidate the mechanism of this cascade reaction, several control experiments are carried out. The initial and crucial step is the selective cleavage of double C-O bonds of the lignin β-O-4 with γ-OH motif with the assistance of the V- catalyst, determining the downstream reaction. Thus treatment of **1d** alone under standard conditions yields 51% of enone (**5a**) and 60% of **4b** (Equation 1, Fig. 3). Then the reaction of **5a** with **2a** affords the desired product **3d** in 45% yield (Equation 2, Fig. 3), indicating that **5a** is a key intermediate. Compared to 61% yield of **3d** in Entry 4 Fig. 2 using **1d** as a starting material, this lower yield is due to **5a** being very unstable and an easily occurring side reactions at the rather high temperature of 140 °C[64], in situ generated enone **5a** from the lignin β-O-4 segment is more efficiently converted to carbazole. Aiming to verify the dehydrogenation

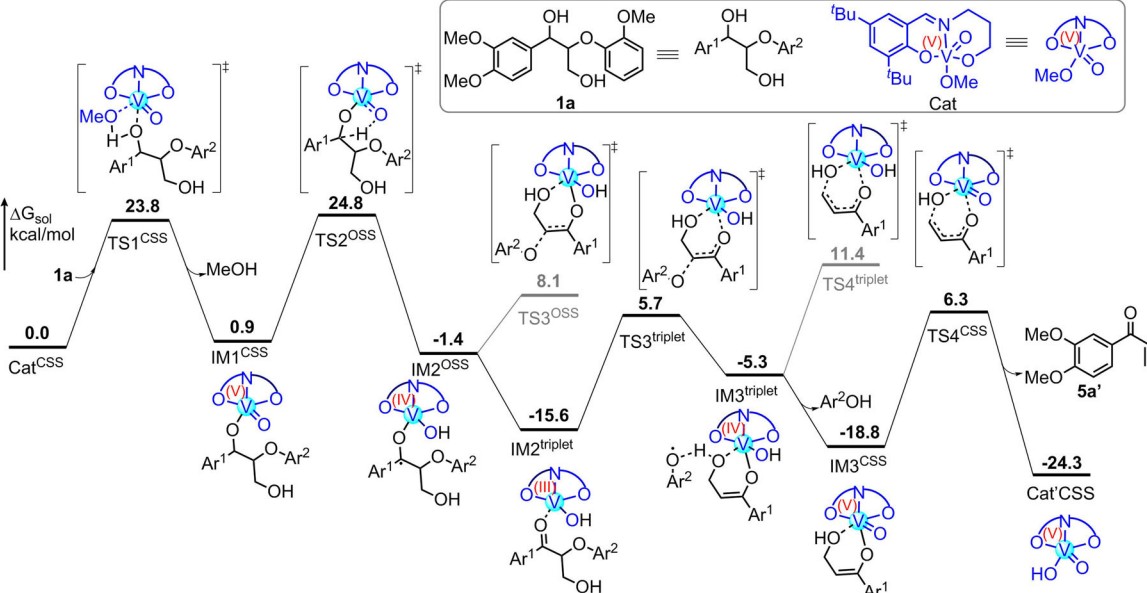

**Fig. 4 | Proposed pathway.** Proposed pathway for the production of carbazoles from lignin β-O-4 motifs ([V] refers to V-complex catalyst).

**Fig. 5 | Computational analysis.** Calculated energy profile of the V-complex catalyzed selective C-O bond cleavage of **1a** (CSS refers to closed-shell singlet and OSS refers to open-shell singlet).

ability of the V-catalyst, the same reaction as given in Equation 2 is performed in the absence of catalyst (Equation 3, Fig. 3). However, only 5% yield of **3d** are obtained (Equation 3, Fig. 3). Similarly, the reactions of **5a** with **2a** with and without V-catalyst are also conducted under argon atmosphere (Equations 4 and 5, Fig. 3), showing that the target product **3d** is formed in 37% yield with V-catalyst, compared with 3% yield of **3d** in the absence of V-catalyst. These results show that the V-catalyst plays an important role for the subsequent intramolecular dehydrogenation. On the basis of these results, a plausible reaction pathway is proposed in Fig. 4: lignin β-O-4 model compounds initially undergo selective V-catalyzed cleavage of the C-O bond to release enones **5** and phenol co-products **4**. Then **5** reacts with **2** via a Diels-Alder cycloaddition to form tetrahydrocarbazoles backbone **6**, which subsequently generates the final product **3** through dehydroaromatization. In the whole reaction the V-complex acts as a bifunctional catalyst for both the selective cleavage of the C-O bond of the lignin β-O-4 model compounds and for the dehydrogenation process.

To gain deeper insight into the detailed reaction mechanism of the vanadium-complex catalyzed selective C-O bond cleavage, density

functional theory (DFT) calculations are performed. The calculated energy profile of the vanadium-catalyzed selective C-O bond cleavage of **1a** is given in Fig. 5 and Supplementary Table 4 as well as Supplementary Data 1.

The reaction is proposed to be initiated by a ligand exchange between the V-catalyst Cat$^{CSS}$ and the benzylic hydroxyl group of **1a** to give intermediate IM1$^{CSS}$ and MeOH, which is agreement with previous work[62]. This process is found to occur via a four-membered transition state TS1$^{CSS}$, with an energy barrier of 23.8 kcal/mol. Then, intermediate IM1$^{CSS}$ undergoes an intramolecular hydrogen transfer via transition state TS2$^{OSS}$ with an energy barrier of 23.9 kcal/mol relative to IM1$^{CSS}$, leading to intermediate IM2$^{OSS}$. The calculations show that the ensuing C(sp$^3$)-O bond cleavage directly from intermediate IM2$^{OSS}$ takes place via transition state TS3$^{OSS}$, which is 9.5 kcal/mol higher in energy than IM2$^{OSS}$. Alternatively, it is found that a spin-crossover phenomenon from a singlet spin state to a triplet spin state can occur to yield the very stable ketone-coordinated V(III) IM2$^{triplet}$, from which the C(sp$^3$)-O bond cleavage via transition state TS3$^{triplet}$ is slightly more favored with an energy barrier of 21.3 kcal/mol relative to IM2$^{triplet}$.

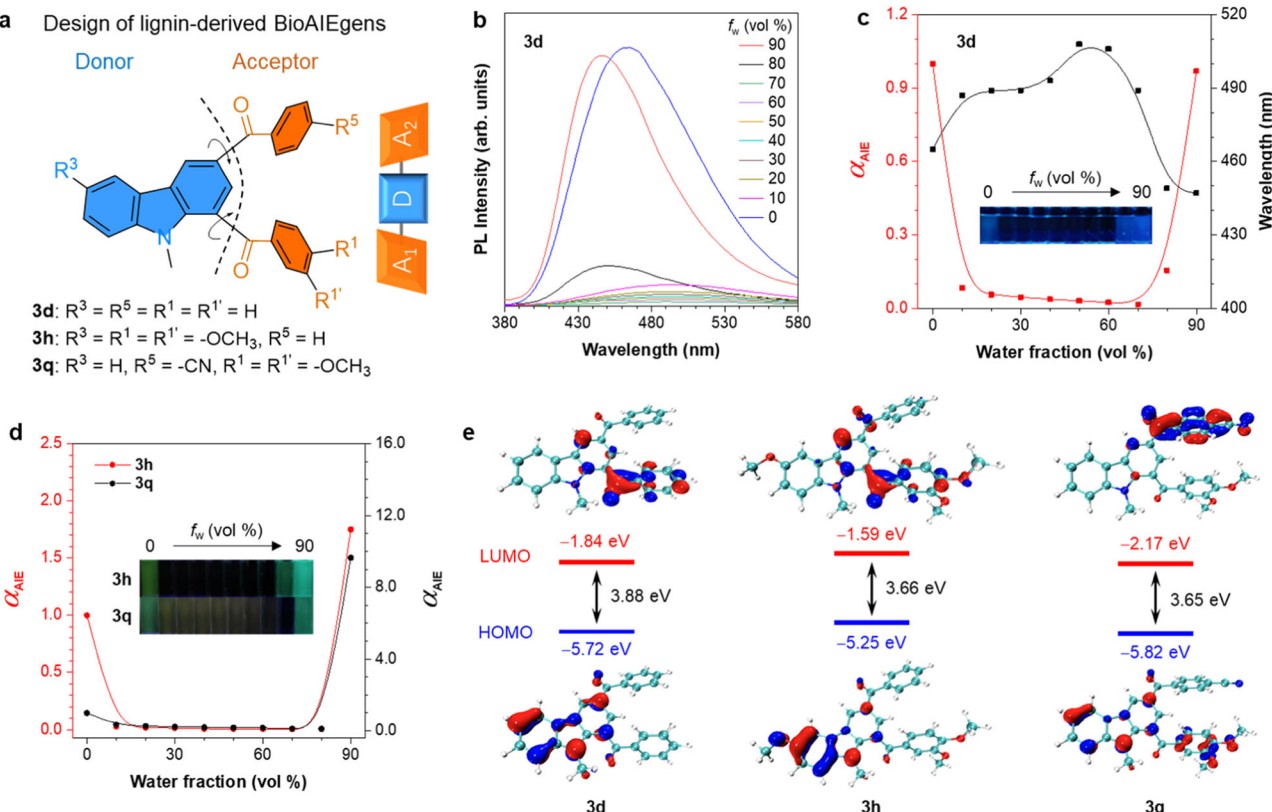

**Fig. 6 | Investigation of optical properties. a** Molecular design of lignin-derived BioAIEgens; **b** PL spectra of **3d** in acetonitrile/water (ACN/H$_2$O) mixtures with different water fractions ($f_w$), $\lambda_{ex}$: 300 nm, concentration: 20 μM; **c** the plots of the $\alpha_{AIE}$ and maximum emission wavelength versus the composition of the aqueous mixture of **3d**, $\alpha_{AIE} = I/I_0$, $I_0 = $ PL intensity in pure ACN, inset: fluorescence photographs taken under 365 nm UV irradiation; **d** plots of $\alpha_{AIE}$ versus the composition of the aqueous mixture of **3h** ($I_0 = $ PL intensity in pure ACN, $\lambda_{ex}$: 311 nm) and **3q** ($I_0 = $ PL intensity in pure THF, $\lambda_{ex}$: 301 nm), inset: fluorescence photographs taken under 365 nm UV irradiation, concentration: 20 μM; **e** frontier molecular orbitals and their corresponding HOMO/LUMO energies of **3d**, **3h**, and **3q**.

Finally, the reaction is completed by a hydroxyl group elimination to deliver enone product **5a'**. The results show that hydroxyl group elimination via a triplet spin state transition state TS4$^{triplet}$ is much higher in energy than that via singlet spin state transition state TS4$^{CSS}$ (11.4 versus 6.3 kcal/mol), showing that the spin-crossover phenomena from triplet spin state (IM3$^{triplet}$) to singlet spin state (IM3$^{CSS}$) has to occur prior to the hydroxyl group elimination. Taken all results together, the hydroxyl group elimination is found to be the rate-determining step of the overall catalytic cycle, with an energy barrier of 25.1 kcal/mol.

## Investigation of optical properties

Additional to establishing a sustainable synthesis of carbazole derivatives their optical properties have been studied as well. It is interesting to note that the carbazole backbones of **3a**–**3v** do possess a chemical structure of two benzoyl moieties as electronic acceptors on the meta positions of the electronic donor of carbazole, constituting a triangular A$_1$-D-A$_2$ configuration (Fig. 6a). The substrates involved herein have four changeable substituents on the A$_1$-D-A$_2$ skeleton, endowing most of them with AIE properties and distinct emission behavior (Supplementary Figs. 36−57). Represented by the non-substituted A-D-A skeleton of **3d** (Fig. 6b, c), the PL intensity decreases immediately when water is added, accompanied by a red-shifted emission that is also observed in a solvent effect experiment with increased solvent polarity (Supplementary Fig. 58), indicating the existence of TICT property[65,66]. When $f_w$ increases up to 80%, the fluorescence revives, which is a typical feature of an AIE property. This might be because the restriction of intramolecular motion in the nanoaggregates leads to an intensified emission (Supplementary Fig. 59). For **3h** and **3q**, the

photophysical properties in the aqueous mixtures show a similar trend as that of **3d**, implying that they have TICT-AIE properties as well (Fig. 6d, Supplementary Figs. 43 and 52). In order to shed more light on these observations, DFT calculations were carried out (Fig. 6e), showing CT effects with an electron shift to the LUMO, which is located either on A$_1$ or on A$_2$, dependent on the substituents. **3d** displays electrons localized on carbazole in the HOMO and an A$_1$ localized LUMO, respectively. Location of the LUMO on A$_1$ may be due to its unsymmetrical A$_1$-D-A$_2$ motif. For **3h**, the substitution of two electron-donating methoxy groups on both D and A$_1$ weakens the electron-acceptor ability of A$_1$, with the LUMO not only localized on A$_1$ but also slightly extending to A$_2$. With regard to **3q**, the LUMO mainly concentrates on A$_2$, owing to the strong electron withdrawing ability of the −CN group. The variable substitutions not only cause varied electronic distributions and locations of HOMO and LUMO, but also lead to different energy differences (band gaps) between the frontier orbitals as exemplified for **3d** (3.88 eV), **3h** (3.66 eV), and **3q** (3.65 eV), matching well with their luminescence colors. These results imply that the electronic distributions can be adjusted by altering the substituents on the A$_1$-D-A$_2$ skeleton, which can help to regulate the photophysical properties of the respective lignin-derived BioAIEgens.

In addition to the fluorescence behavior in solution and aggregate states, the photophysical properties in the solid state have been further studied. With the aim of investigating the structure-property relationships of solid-state fluorescence materials, single crystals of **3p** and **3q** have been obtained for further analysis. Similar to the solution and aggregate states, the unsubstituted solid state of **3d** still exhibits blue-violet fluorescence with a wavelength of 436 nm. **3p** and **3q** show a significant red-shift (**3p**: 493 nm; **3q**: 511 nm) compared to **3d**, probably

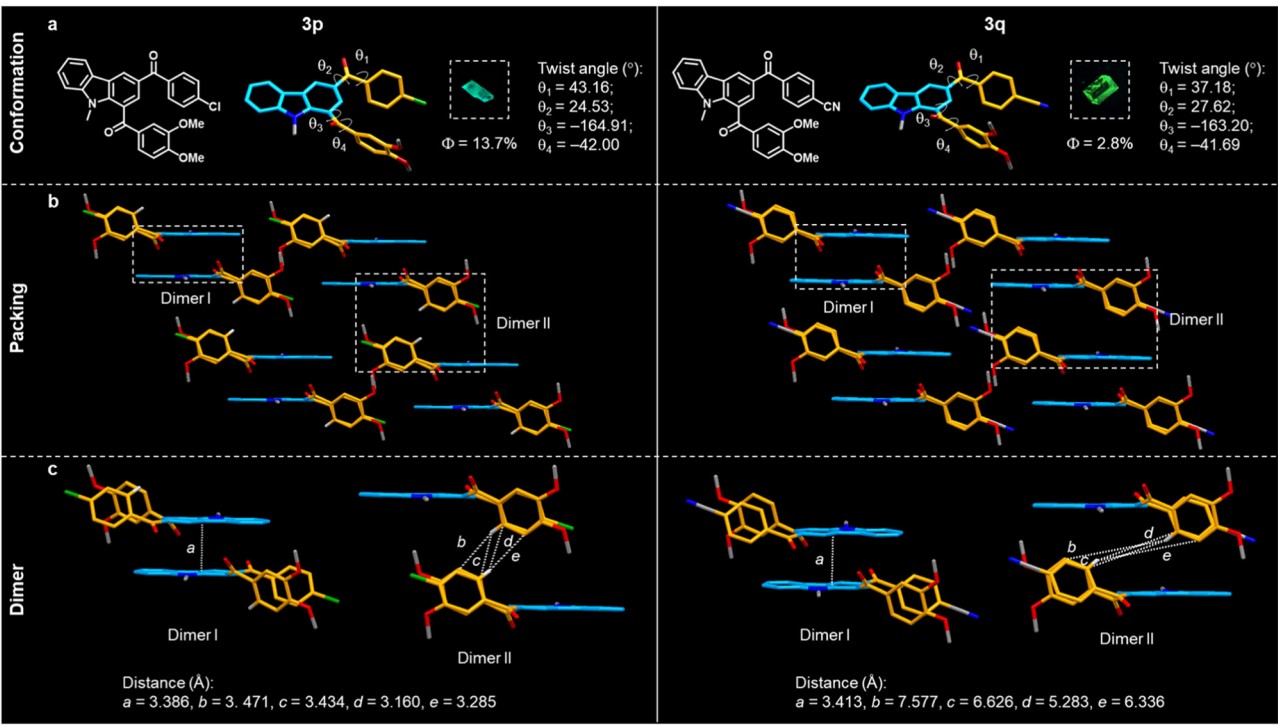

**Fig. 7 | The crystal structures of** 3p and 3q. **a** Molecular conformation of **3p** and **3q**; **b** packing of **3p** and **3q**; **c** respective dimer structures of **3p** and **3q**.

due to the presence of electronically different substituents[67,68], thus enhancing the characteristics of CT effects. With regard to the emission intensity, the solid-state quantum yield values of **3d**, **3p**, and **3q** are 3.5%, 13.7% and 2.8%, respectively, and these trends are identical to the emission intensity reported in the solid-state emission spectra (Supplementary Fig. 60). In order to uncover the causes for the differences in emission intensity, the crystal structures of **3p** and **3q** have been analyzed in detail (Fig. 7 and Supplementary Table 5). The results show that the torsion angles at four comparable positions of **3p** (43.16°, 24.53°, −164.91°, −42.00°) and **3q** (37.18°, 27.62°, −163.20°, −41.69°) are quite close to each other, indicating similar molecular conformations (Fig. 7a). In addition, the packing modes of **3p** and **3q** are similar as well, giving two types of dimer modes (dimers I and II) (Fig. 7b). Further analysis demonstrates that they display slight differences in the fundamental structures of the two types of dimers (Fig. 7c). Specifically, in dimer I, the interplanar distances between the carbazole planes of **3p** (3.386 Å) and **3q** (3.413 Å) are quite similar. However, in dimer II, the four benzene rings between the two molecules constituting the dimer II in **3p** exhibit intermolecular C···H distances from 3.160 to 3.471 Å. The distances between the four benzene rings in **3q** are quite long (5.283–7.577 Å), exceeding the normal intermolecular interactions, giving rise to the weaker solid-state emission of **3q**[69,70]. The above described results suggest that in the case of very similar conformations subtle differences between dimers in the packing structures can nevertheless play a prominent role for the observed emission intensities.

In summary, a vanadium-catalyzed direct transformation of phenolic β-O-4 lignin segments, containing γ-OH with 3-alkenylated indoles has been developed for the construction of carbazole-based bioAIEgens without an external H/O resource. A selective tandem cleavage of two C-O bonds /Diels–Alder cycloaddition/dehydrogenative aromatization sequence is established as the reaction pathway, both through experimental studies and DFT calculations. As intended, the lignin-derived carbazoles with A-D-A configurations exhibit TICT-AIE performance, endowing the lignin-depolymerized derivatives with unique photophysical properties. The present protocol provides a concise and sustainable route to functionalized carbazoles with AIE properties from renewable substrates, creating a bridge between lignin and BioAIEgens.

## Methods

### Typical procedure for carbazole derivatives synthesis from lignin β-O-4 model compounds

Quantification of phenol derivatives: lignin model compound **1** (0.4 mmol), alkenylated indole **2** (0.2 mmol), catalyst (10 mol%) and toluene (4 mL) are added into a pressure tube (35 mL). The mixture was sealed and heated to 140 °C for 20 h. After reaction, the solution is cooled to room temperature, and diluted to 25 mL volumetric flask with DMF. The phenol derivatives are analyzed by HPLC using an external standard calibration curve method. Quantification of carbazole derivatives: lignin model compound **1** (0.4 mmol), alkenylated indole **2** (0.2 mmol), catalyst (10 mol%) and toluene (4 mL) are added into a pressure tube (35 mL). The mixture is sealed and heated to 140 °C for 20 h. After reaction, the solution is cooled to room temperature. The carbazole products are purified by column chromatography using petroleum ether/ethyl acetate (3:1) to give the isolated yield.

### Computational details

All the calculations are performed at the (u)B3LYP-D3(BJ)[71–73] level of theory using Gaussian 09 package[74]. The geometry optimizations are carried out with a mixed basis set of LANL2TZ(f) for V and 6-31 G(d,p) for other atoms. Frequencies are computed analytically at the same level of theory to confirm whether the structures are minima (no imaginary frequencies) or transition states (only one imaginary frequency). Selected transition-state structures are confirmed to connect the correct reactants and products by intrinsic reaction coordinate calculations[75,76]. To obtain better accuracy, energies for the optimized geometries were recalculated using the solution-phase single-point calculations with a larger basis set, which is LANL2TZ(f) for V and 6-311 + G(d,p) for all other atoms. Solvation effects (solvent = toluene, ε = 2.374) are taken into account by performing single-point calculations with the SMD

model[77]. The final free energies reported in the article are the large basis set single-point energies corrected by gas-phase Gibbs free energy correction (at 298.15 K).

## Data availability

All relevant data are available within the article as well as its Supplementary Information and from the corresponding authors on request. Crystallographic data for structures **3d**, **3p** and **3q** reported in this article have been deposited at the Cambridge Crystallographic Data Centre, under deposition numbers CCDC 2212869, 2212868 and 2212870, respectively. These data can be obtained free of charge from The Cambridge Crystallographic Data Centre via https://www.ccdc.cam.ac.uk/structures/.

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

## Acknowledgements

Support from the National Natural Science Foundation of China (22078317, 22108272, 21601087, 22073066), Natural Science Foundation of Jiangsu Province (BK20231296), the science and technology bureau of Dalian city (No. 2021RT04) and Technical University of Munich.

## Author contributions

T.Z., B.Z. and X.-M.C. conceived the study and directed the project; T.G. and B.Z. designed and performed the experiments; W.Z. and D.X. performed the control experiments; Y.L., X.Z. and X.-M.C. performed

photophysical investigation for AIEgens; D.P. and G.H. performed the DFT calculations. H.W. performed a part of HPLC measurements. B.Z., X.-M.C. and G.H. wrote the manuscript. F.E.K. and T.Z. improved the manuscript. All the authors discussed the results and commented on the manuscript.

## Competing interests

The authors declare no competing interests.
