## [Peer Review File · Nature Communications]

Towards BioAIEgens from lignin β -O-4 motifs as renewable resourcesREVIEWER COMMENTS

Reviewer #1 (Remarks to the Author):

This manuscript by Zhang and co-workers introduced de-polymerization of lignin β -O-4 segments for vanadium-catalyzed benzannulation reactions with 3-alkenylated indole moieties under redox-neutral conditions for the construction of functionalized carbazole derivatives. In depth DFT calculations confer detailed mechanistic analysis on the reaction pathways which includes selective cleavage of double C-O bonds/Diels-Alder cycloaddition followed by vanadium promoted dehydrogenative aromatization reaction sequences. In addition, photo-physical characterizations of the A-D-A type functionalized carbazoles revealed their aggregation-induced emission (AIE) with twisted intramolecular charge transfer properties. With respect to novelty, in the synthesis part, this paper utilized two reported strategies, namely, generation of enone and its application for the [4+2] Diels-Alder reaction, which eventually provide carbazole after oxidation. The well established vanadium based catalytic system is employed in oxidation of biomass Lignin model to generate enone.

Homogeneous vanadium-based catalyst with Schiff base ligands was previously introduced and well explored for redox-neutral C-O bond cleavage over benzylic oxidation of Lignin model compounds (Angew. Chem. Int. Ed. 2010, 49, 3791–3794). Similar catalytic approach has been utilized here for synthesis of the intermediate enone 5a which efficiently converted to carbazole by reacting with alkenyl indole moieties.

As A-D-A analogue, synthesis of di-benzoyl substituted carbazole derivatives were previously documented in several papers employing other transition metal catalysts (Chem. Sci. 2013, 4, 3416–3420; Org. Chem. Front. 2014, 1, 707–711; J. Org. Chem. 2020, 85, 14, 9117–9128) which degrade the synthetic novelty of the synthesized compounds tabulated herein.

This paper does not meet both criteria-novelty and very high level of specialist interest-for publication in Nature Communications.

Few observations are noted below:

- (1) The authors have used “bio-carbazole” terminology for the synthesized compounds in manuscript. Unnecessary implementation of the term “bio-” does not meet any special criteria of the carbazole analogues. This terminology should have been removed throughout the manuscript.
- (2) Authors should provide a detailed information on optimization by varying mol% of the catalyst along with the equivalent of prototypes involved in the reaction scenario.
- (3) All the reactions have been performed with N-protected alkenyl indole derivatives. What will the outcome of the reaction when N-H free indole have been introduced?
- (4) There are several formatting mistakes present in the reference section of the manuscript. Authors should recheck and rectify the errors.

Reviewer #2 (Remarks to the Author):

This work presented the results on the production of carbazoles from a lignin β -O-4 segments, addressing the synthesis, mechanism and properties. This article is of significant important for bridging the gap between lignin utilization and Bio-AIEgens. Starting from model compounds mimicking lignin main unit, the authors designed cascade reactions to produce carbazole derivatives in good to excellent yield. The mechanism of the reaction is partially studied experimentally and thoroughly supported by the DFT calculations. Moreover, the lignin-derived carbazoles with A-D-A configurations exhibit TICT-AIE performance, and the mechanism of luminescence is studied as well. This study is related to the hot field of lignin utilization, and the transformation of lignin models to value-added carbazoles with AIE performance is of highly novelty and important. This is a challenging topic and it is well addressed by the authors. I recommend the publication of this contribution in Nature Commun. after minor modifications.

1. In scheme 2, the authors mentioned that V-catalyst plays an important role for the intramolecular dehydrogenation on the basis of the results of Equations 2-3 under air condition. Do the authors consider these reactions under argon environment? More comparative experiments under argon with and without V-catalyst may further illustrate the dehydrogenation role of V-catalyst.
2. Do the authors try the commercial V-based catalyst such as VO(OEt)₃ as catalyst with easily available 2,2'-Bipyridine ligand? I recommend that the authors give this data for comparison.
3. As the authors claimed, the lignin model material used for the reaction was obtained from the synthesis, so is it possible to produce a dimer of β -O-4 linkages containing a γ -OH group obtained from real lignin degradation in your catalytic systems? Have the authors tried the realistic lignin?
4. A test to confirm the generation of tetrahydrocarbazoles backbone 6 could be added to improve the quality of this work, although it is not required.
5. In the abstract, the authors illustrate that this reaction is redox-neutral condition. This reaction carried out under air condition, and oxygen in air promotes to dehydrogenation. In my opinion, the oxygen could be considered as an oxidant in the reaction. So the redox-neutral condition is not rigorous for this reaction. I suggest that the authors delete it.
6. Some figures appear in a low resolution, please replace them in a clearer style. Figure 2 and Figure 3 are so unclear that it is not possible to see the details in the figure. The specifications in Figure 1 are very small and the pictures must be enlarged to see the details.
7. In the supporting information Page 4, please delete "derivative".

Reviewer #3 (Remarks to the Author):

Zhang and co-workers report an investigation on synthesis of carbazole-based BioAIEgens from lignin dimers, and the mechanism and properties of carbazoles products were addressed. This work is interesting in the sense it opens an interesting route to aminated products from lignin, an important topic for which innovation is expected by the chemical industry. The synthetic protocol is efficient, a one-pot cascade reaction without additional oxidant/reductant sources. The photophysical results

present a new class of bioresource-derived BioAIEgens with TICT behaviors depending on the unique A-D-A structures. This work not only provides a new class of lignin-derived BioAIEgens via an interesting synthetic route, but also paves the potential way to the value-adding of lignin-derived biochemicals. Considering these results, I support this paper to be published in Nature Commun. after minor modification.

1. The vanadium with tridentate Schiff base ligand was synthesized. How is about commercial V-based catalyst? Please the authors give these results for comparison.
2. The article states that the molecular design is based on an A-D-A structure. What are the advantages of this structure compared to D-A and D-A-D structures?
3. The authors used the term of "BioAIEgen" in the title. Please explain and further elaborate the advantages of such BioAIEgen in the manuscript, which is very important to help the readers to understand the superiority of the lignin-derived BioAIEgens.
4. Scheme 1 and Fig. 2a use R1 and R2 as variable substituents in their chemical structures, but not consistent. It is necessary to unify them for clear discussion.
5. Fig. 2 discusses the photophysical properties of 3d, 3h, and 3q at molecular state, while Fig. 3 discusses the aggregate structures of 3p and 3q at solid state. However, the discussions are in one section without clear discrimination. Please clarify this point.
6. Carbazole is commonly used to construct molecules with phosphorescent properties. Has the author explored the phosphorescence?
7. Please proof the references. The journal titles are not abbreviated for uniform, such as Refs. 16 and 36.

Response to the comments of the reviewers (NCOMMS-23-18138)

Towards BioAIEgens from lignin β -O-4 motifs as renewable resources: synthesis, mechanism, properties

Tenglong Guo, Yuting Lin, Deng Pan, Xuedan Zhang, Wenqing Zhu, Xu-Min Cai, Genping Huang, Hua Wang, Dezhu Xu, Fritz E. Kühn, Bo Zhang, and Tao Zhang

Dear reviewers,

Thank you for the time and effort you have spent in reviewing our manuscript. We particularly appreciate the insightful and constructive comments. We carefully revised and improved our manuscript following the comments and suggestions. Detailed replies and comments are listed below in a point by point fashion. All the changes are highlighted in the revised manuscript.

Reviewer #1 (Remarks to the Author):

This manuscript by Zhang and co-workers introduced de-polymerization of lignin β -O-4 segments for vanadium-catalyzed benzannulation reactions with 3-alkenylated indole moieties under redox-neutral conditions for the construction of functionalized carbazole derivatives. In depth DFT calculations confer detailed mechanistic analysis on the reaction pathways which includes selective cleavage of double C-O bonds/Diels-Alder cycloaddition followed by vanadium promoted dehydrogenative aromatization reaction sequences. In addition, photo-physical characterizations of the A-D-A type functionalized carbazoles revealed their aggregation-induced emission (AIE) with twisted intramolecular charge transfer properties. With respect to novelty, in the synthesis part, this paper utilized two reported strategies, namely, generation of enone and its application for the [4+2] Diels-Alder reaction, which eventually provide carbazole after oxidation. The well established vanadium based catalytic system is employed in oxidation of biomass Lignin model to generate enone.

Homogeneous vanadium-based catalyst with Schiff base ligands was previously introduced and well explored for redox-neutral C-O bond cleavage over benzylic oxidation of Lignin model compounds (Angew. Chem. Int. Ed. 2010, 49, 3791-3794). Similar catalytic approach has been

utilized here for synthesis of the intermediate enone 5a which efficiently converted to carbazole by reacting with alkenyl indole moieties.

As A-D-A analogue, synthesis of di-benzoyl substituted carbazole derivatives were previously documented in several papers employing other transition metal catalysts (*Chem. Sci.* 2013, 4, 3416 – 3420; *Org. Chem. Front.* 2014, 1, 707-711; *J. Org. Chem.* 2020, 85, 14, 9117-9128) which degrade the synthetic novelty of the synthesized compounds tabulated herein.

This paper does not meet both criteria-novelty and very high level of specialist interest-for publication in *Nature Communications*.

Author reply: Generally, one-pot reaction coupling of several reactions still remains a challenge due to the incompatible catalysis systems, which would have to be applied for the different reaction steps. In the reaction system presented here, carbazole synthesis includes the selective cleavage of C-O bonds, Diels-Alder cycloaddition and dehydrogenative aromatization. Therefore, a catalyst system is needed that enables all three reactions in a one-pot fashion. Although homogeneous vanadium-based catalysts with Schiff base ligands were previously introduced to generate enones from lignin model compounds (*Angew. Chem. Int. Ed.* 2010, 49, 3791 – 3794), **the subsequent reactions of enones with alkenyl indole moieties to form carbazoles generally require an oxidant or catalyst as well, to achieve dehydrogenation. In this work, the V-complex was found to act as a bifunctional catalyst for both the selective cleavage of the C-O bond of the lignin β -O-4 model compounds and the dehydrogenation process, which is crucial for this reaction and has not been reported in previous work.** The dehydrogenation catalyst capability of the V-complex is proven by control experiments, and shown in Fig. 3 and Equations 4-5 on Page 23 in the revised manuscript. For the synthesis of carbazoles, so far extensive carbazoles synthetic strategies focused on intermolecular cross-coupling reactions between C-H/C-X bonds (X = halo, N, O, C, etc), oxidative intramolecular C-H/C-H cross coupling of prefunctionalized diarylamines, or the construction of a benzene ring onto substituted indoles (*Chem. Sci.* 2013, 4, 3416–3420; *Org. Chem. Front.* 2014, 1, 707-711; *J. Org. Chem.* 2020, 85(14), 9117-9128). However, most synthetic strategies utilize expensive noble metal catalyst, excess oxidants and non-renewable substrates, which raises both atom economy and environmental concerns. Therefore, **this work features lignin β -O-4 motifs with γ -OH functionalities as**

renewable substrates, without expensive metal catalysts and excess oxidants, highly coupled multi-step transformation, and a sustainable universal approach, providing a concise and sustainable route to functional carbazoles from renewable biomass. This approach is novel and has not been reported before.

N-participated lignin depolymerization is of significant importance and an indispensable asset for biomass conversion in order to afford heterocyclic aromatic compounds. Its complex structure makes it difficult to controllably degrade lignin, limiting further application. However, most of the target N-heterocyclic aromatics are so far confined to applications as medicinal intermediates, large volume application is prohibited by the need of further functionalization. Therefore, designing lignin-derivatized BioAIEgens could bring new aspects to the research on lignin transformation, widening lignin application areas and expanding the product pool of biomass conversion to meet further biorefinery demands. Although the synthesis of di-benzoyl substituted carbazole derivatives was previously documented in several papers, **the photophysical properties of these compounds were not investigated.** In this work, the photophysical properties of these carbazole products have been studied in some detail as a new class of BioAIEgens with twisted intramolecular charge transfer. Distinctions of emission behavior were also revealed based on unique acceptor-donor-acceptor-type molecular conformations as well as molecular packings. **All these results are important supplements for lignin-based BioAIE materials. Based on these facts, our work expands potential applications of lignin-based chemicals and builds a bridge between lignin and BioAIE materials, which is of high importance to meet value-added biorefinery demands.**

Therefore, our work is original. The major features are as follows:

1. A successful strategy for the direct transformation of lignin β -O-4 model compounds with γ -OH moieties into carbazoles in high yields, without an additional hydrogen/oxidation steps has been developed and successfully executed for the first time.
2. The applied V-based catalyst displays outstanding performance for the transformation and enables a selective cascade cleavage of C-O bonds/Diels–Alder cycloaddition/dehydrogenative aromatization.
3. The establishment of the triangular A-D-A skeleton based on the electron donor capability of carbazole and the corresponding electron acceptor ability of benzoyl results

in the intended intramolecular charge transfer (TICT) and AIE properties.

4. The concept of “lignin-based BioAIEgen” is demonstrated, thus creating a bridge between lignin and BioAIE materials and allowing fluorescence applications.

This work opens an energy-efficient route for the direct conversion of lignin β -O-4 with γ -OH moieties into valuable carbazoles, providing a concise and sustainable route to functional carbazoles with AIE properties. We are convinced that the work described here is of significant importance for both basic and applied chemistry.

The novelty and importance of this work have also been approved by the other two reviewers. Reviewer #2 commented that “This article is of significant important for bridging the gap between lignin utilization and Bio-AIEgens. ... This study is related to the hot field of lignin utilization, and the transformation of lignin models to value-added carbazoles with AIE performance is of highly novelty and important. This is a challenging topic and it is well addressed by the authors.”

Reviewer #3 also commented that “This work is interesting in the sense it opens an interesting route to aminated products from lignin, an important topic for which innovation is expected by the chemical industry. The synthetic protocol is efficient, a one-pot cascade reaction without additional oxidant/reductant sources. ... This work not only provides a new class of lignin-derived BioAIEgens via an interesting synthetic route, but also paves the potential way to the value-adding of lignin-derived biochemicals.”

We hope that referee #1 can, based on these clarifications, appreciate and approve the novelty of the work presented in this manuscript.

Question 1: The authors have used “bio-carbazole” terminology for the synthesized compounds in manuscript. Unnecessary implementation of the term “bio-” does not meet any special criteria of the carbazole analogues. This terminology should have been removed throughout the manuscript.

Author reply: As suggested, all of term “bio-” before carbazole has been removed throughout the manuscript.

Question 2: Authors should provide a detailed information on optimization by varying mol% of the catalyst along with the equivalent of prototypes involved in the reaction scenario.

Author reply: As suggested, we have added the results with variation of catalyst loading (mol%) and variation of substrate stoichiometry to the revised supporting information on Page S6. The corresponding discussion has been added into the revised manuscript on Page 6. It reads “Based on this exciting reactivity, the experiment parameters (catalyst loading, substrates ratio, and reaction temperature) were optimized (Supplementary Tables 1-3).”

Supplementary Table R1. Synthesis of carbazole derivatives **3a** under variation of catalyst loading.^a

Entry	Cat. Loading (mol%)	3a Yield [%] ^b	4a Yield [%] ^b
1	1	16	28
2	5	28	30
3	10	89	99
4	15	70	64

^a Reaction conditions: **1a** (0.2 mmol), **2a** (0.1 mmol), V-based complex (x mol%), toluene (2 mL) in air, reaction time (t) = 20 h, the yields of **3a** and **4a** are calculated based on the amounts of **2a** and **1a**, respectively; ^b Yields are determined by HPLC with an external standard method.

Supplementary Table R2. Synthesis of carbazole derivatives **3a** under variation of substrates stoichiometry.^a

Entry	1a (mmol)	2a (mmol)	1a:2a	3a Yield [%] ^b	4a Yield [%] ^b
1	0.1	0.1	1:1	48	79
2	0.2	0.1	2:1	89	99
3	0.1	0.2	1:2	42	83

^a Reaction conditions: V-based complex (10 mol%), toluene (2 mL) in air, reaction time (t) = 20 h, the yields of **3a** and **4a** are calculated based on the amounts of **2a** and **1a**, respectively; ^b Yields are determined by HPLC with an external standard method.

Question 3: All the reactions have been performed with N-protected alkenyl indole derivatives. What will the outcome of the reaction when N-H free indole have been introduced?

Author reply: As suggested, the protocol has been extended to the reaction of a N-H free indole. The results show that such a reaction is complicated, leading to a number of products. We isolated the main product N-(3-oxo-3-phenylpropyl)-substituted product in 21% yield (Equation 1, Fig R1). NMR spectra are shown below in Fig. R2. No desired carbazole product is formed. This is mainly because nucleophilic addition between N-H free indole and enone generated from the lignin model compound is preferred, compared to a Diels-Alder reaction. In order to obtain high selectivity of carbazole products, N-protected alkenyl indole derivatives are used as substrates.

Fig. R1. Control experiment

(E)-3-(1-(3-oxo-3-phenylpropyl)-1H-indol-3-yl)-1-phenylprop-2-en-1-one: Yield 21%. ^1H NMR (400 MHz, CDCl_3) δ 8.07 (d, $J = 15.5$ Hz, 2 H, $\text{CH}=\text{CHCOPh}$), 8.02 (m, 2 H, aromatic CH), 7.90 (m, 2 H, aromatic CH), 7.65 (s, 1 H, 2-H of indolyl), 7.52 (m, 5 H, aromatic CH), 7.44 (t, 3 H, aromatic CH), 7.33 (m, 2 H, aromatic CH), 4.66 (t, 2 H, $\text{NCH}_2\text{CH}_2\text{COPh}$), 3.52 (t, 2 H, $\text{NCH}_2\text{CH}_2\text{COPh}$). $^{13}\text{C}\{^1\text{H}\}$ NMR (100 MHz, CDCl_3) δ 197.1 and 190.8 (Cq, $\text{C}=\text{O}$), 139.2, 137.3, 136.2, 126.6, and 113.4 (Cq), 138.7, 134.2, 133.8, 132.2, 128.9, 128.6, 128.4, 128.1, 123.4, 121.8, 121.1, 117.5, and 110.2 (CH), 41.4 ($\text{NCH}_2\text{CH}_2\text{COPh}$), 38.4 ($\text{NCH}_2\text{CH}_2\text{COPh}$). HRMS Calcd for $\text{C}_{26}\text{H}_{21}\text{NO}_2$ $[\text{M}]^+$: 379.1572; Found: 379.1564.

Fig. R2 NMR spectra

Question 4: There are several formatting mistakes present in the reference section of the manuscript. Authors should recheck and rectify the errors.

Author reply: We appreciate the reviewer's comment. As suggested, we have rechecked and rectified the errors in the reference section of the manuscript.

Reviewer #2 (Publish after minor modifications):

This work presented the results on the production of carbazoles from a lignin β -O-4 segments, addressing the synthesis, mechanism and properties. This article is of significant important for bridging the gap between lignin utilization and Bio-AIEgens. Starting from model compounds mimicking lignin main unit, the authors designed cascade reactions to produce carbazole derivatives in good to excellent yield. The mechanism of the reaction is partially studied experimentally and thoroughly supported by the DFT calculations. Moreover, the lignin-derived carbazoles with A-D-A configurations exhibit TICT-AIE performance, and the mechanism of luminescence is studied as well. This study is related to the hot field of lignin utilization, and the transformation of lignin models to value-added carbazoles with AIE performance is of highly novelty and important. This is a challenging topic and it is well addressed by the authors. I recommend the publication of this contribution in Nature Commun. after minor modifications.

Question 1: In scheme 2, the authors mentioned that V-catalyst plays an important role for the intramolecular dehydrogenation on the basis of the results of Equations 2-3 under air condition. Do the authors consider these reactions under argon environment? More comparative experiments under argon with and without V-catalyst may further illustrate the dehydrogenation role of V-catalyst.

Author reply: As suggested, the reaction of **5a** with **2a** with and without V-catalyst have been conducted under argon atmosphere. The results show that the target product **3d** is formed in 37% yield with V-catalyst, compared to only 3% yield of **3d** without V-catalyst in Fig. 3R. These results suggest that the V-catalyst plays an important role for the subsequent intramolecular dehydrogenation. We have added the corresponding results and discussion in the revised manuscript on Page 8. Equations were added to Fig. 3 as equations 4 and 5. The text in the revised manuscript now reads: “Similarly, the reactions of **5a** with **2a** with and without V-catalyst were also conducted under argon atmosphere (Equations 4 and 5, Fig. 3), showing that the target product **3d** was formed in 37% yield with V-catalyst, compared with 3% yield of **3d** in the absence of V-catalyst.”

Fig.R3 Control Experiments

Question 2: Do the authors try the commercial V-based catalyst such as $\text{VO}(\text{OEt})_3$ as catalyst with easily available 2,2'-Bipyridine ligand? I recommend that the authors give this data for comparison.

Author reply: As suggested, the model reaction of **1a** with **2a** using 10 mol% $\text{VO}(\text{OEt})_3$ as catalyst and 15 mol% 2,2'-bipyridine as ligand has been carried out, affording carbazole **3a** in 55% yield along with 36% guaiacol **4a**. This result has been added into Supplementary Table 1 as entry 9 and the discussion was added in the revised manuscript on Page 6. It now reads: “In contrast, changing the catalyst to commercial $\text{VO}(\text{OEt})_3$ and 2,2'-bipyridine ligand, the carbazole **3a** is formed only in 55% yield (Supplementary Table 1, entry 9).”

Question 3: As the authors claimed, the lignin model material used for the reaction was obtained from the synthesis, so is it possible to produce a dimer of β -O-4 linkages containing a γ -OH group obtained from real lignin degradation in your catalytic systems? Have the authors tried the realistic

lignin?

Author reply: As suggested, we have tried the realistic lignin in our catalytic systems. Unfortunately, only a small amount of guaiacol can be detected after reaction and no β -O-4 linkages containing a γ -OH group. During conversion of lignin feedstocks, β -O-4 linkages containing a γ -OH group are very difficult to retain, since cleavage of C-O bonds in β -O-4 linkages containing a γ -OH group easily occurs.

Question 4: A test to confirm the generation of tetrahydrocarbazoles backbone **6** could be added to improve the quality of this work, although it is not required.

Author reply: In order to avoid oxidants, we have performed the reaction of **5a** with **2a** without V-catalyst under argon atmosphere to capture tetrahydrocarbazole intermediate (Fig. 3, equation 5). However, only 3% yield of **3d** is detected and no tetrahydrocarbazole intermediate can be obtained. We assume that the tetrahydrocarbazole intermediate might be unstable and is transformed to carbazole once generated.

Question 5: In the abstract, the authors illustrate that this reaction is redox-neutral condition. This reaction carried out under air condition, and oxygen in air promotes to dehydrogenation. In my opinion, the oxygen could be considered as an oxidant in the reaction. So the redox-neutral condition is not rigorous for this reaction. I suggest that the authors delete it.

Author reply: Thank you for the helpful comment. As suggested, we have deleted the term “redox-neutral” in the abstract and manuscript.

Question 6: Some figures appear in a low resolution, please replace them in a clearer style. Figure 2 and Figure 3 are so unclear that it is not possible to see the details in the figure. The specifications in Figure 1 are very small and the pictures must be enlarged to see the details.

Author reply: Thank you for the helpful comment. As suggested, we have replaced Figs. 2 and 3 in a clearer style as Figs. 6 and 7 in the revised manuscript. Additionally, Fig. 1 has also been enlarged as Fig. 5 in the revised manuscript. The Figs. are shown as below.

Fig.R4 Calculated energy profile of the V-complex catalyzed selective C-O bond cleavage of **1a** (CSS refers to closed-shell singlet and OSS refers to open-shell singlet).

Fig.R5 Investigation of optical properties. **a** Molecular design of lignin-derived BioAIEgens. **b** PL spectra of **3d** in acetonitrile/water (ACN/H₂O) mixtures with different f_w . λ_{ex} : 300 nm, concentration: 20 μ M. **c** The plots of the α_{AIE} and maximum emission wavelength versus the composition of the aqueous mixture of **3d**, $\alpha_{AIE} = I/I_0$, I_0 = PL intensity in pure ACN. Inset: fluorescence photographs taken under 365 nm UV irradiation, concentration: 20 μ M. **d** Plots of α_{AIE} versus the composition of the aqueous mixture of **3h** (I_0 = PL intensity in pure ACN, λ_{ex} : 311 nm) and **3q** (I_0 = PL intensity in pure THF, λ_{ex} : 301 nm). Inset: fluorescence photographs taken under 365 nm UV irradiation, concentration: 20 μ M. **e** Frontier molecular orbitals and their corresponding HOMO/LUMO energies of **3d**, **3h**, and **3q**.

Fig. R6 The crystal structures of **3p** and **3q**. **a** Molecular conformation of **3p** and **3q**. **b** Packing of **3p** and **3q**. **c** Respective dimer structures of **3p** and **3q**.

Question 7: In the supporting information Page 4, please delete “derivative”.

Author reply: We have deleted “derivative” in the supporting information on Page 4 as suggested.

Reviewer #3 (Publish after minor modifications):

Zhang and co-workers report an investigation on synthesis of carbazole-based BioAIEgens from lignin dimers, and the mechanism and properties of carbazoles products were addressed. This work is interesting in the sense it opens an interesting route to aminated products from lignin, an important topic for which innovation is expected by the chemical industry. The synthetic protocol is efficient, a one-pot cascade reaction without additional oxidant/reductant sources. The photophysical results present a new class of bioresource-derived BioAIEgens with TICT behaviors depending on the unique A-D-A structures. This work not only provides a new class of lignin-derived BioAIEgens via an interesting synthetic route, but also paves the potential way to the value-adding of lignin-derived biochemicals. Considering these results, I support this paper to be published in Nature Commun. after minor modification.

Question 1: The vanadium with tridentate Schiff base ligand was synthesized. How is about commercial V-based catalyst? Please the authors give these results for comparison.

Author reply: The model reaction of **1a** with **2a** using 10 mol% VO(OEt) as catalyst and 15 mol% 2,2'-bipyridine as ligand has been carried out, affording the carbazole **3a** in 55% yield along with 36% guaiacol **4a**. The result has been added to the Supplementary Table 1 as entry 10 and the discussion has been added in the revised manuscript on Page 6. It now reads: “In contrast, changing the catalyst to commercial VO(OEt)₃ and 2,2'-bipyridine ligand, the carbazole **3a** is formed only in 55% yield (Supplementary Table 1, entry 9).”

Question 2: The article states that the molecular design is based on an A-D-A structure. What are the advantages of this structure compared to D-A and D-A-D structures?

Author reply: Molecules with a D-A architecture have been recognized to possess high PLQY and high light-harvesting capability owing to their narrow optical band gaps (*Angew. Chem. Int. Ed.*, **2023**, e202303476). Meanwhile, the easily controllable charge transfer (CT) excited states of D-A compounds are widely considered as an effective method to manipulate the properties of luminescent materials, and thus extensively employed in the field of optical materials (*Adv. Healthcare Mater.*, **2022**, *11*, 2201158). The A-D-A and D-A-D molecules are two specific types of the D-A structures. Molecules with D-A-D structure typically exhibit strong CT processes and small singlet-triplet energy gap, which facilitate inter-system crossing and reverse intersystem crossing. Therefore, they are extensively utilized in applications such as thermally activated delayed fluorescence materials and room-temperature phosphorescent materials (*Nat. Commun.*, **2017**, *8*, 14987; *Nature Photon.*, **2018**, *12*, 235-240; *Angew. Chem. Int. Ed.*, **2019**, *58*, 16445-16450). In recent years, A-D-A molecules have gained increasing attention and have been extensively applied in the field of organic photovoltaics (*Chem. Soc. Rev.*, **2020**, *49*, 2828). Although research and reports on A-D-A molecules are relatively limited compared to those with the D-A-D structure, some reports suggest that A-D-A molecules can exhibit stronger CT effects and light-harvesting capability compared to D-A-D structure. These molecules also hold great potential in biomedical applications (*Angew. Chem. Int. Ed.*, **2023**, e202303476; *Small*, **2021**, *17*, 2102044; *ACS Nano*, **2019**, *13*, 12901–12911). Therefore, studying the photophysical properties of organic molecules with the A-D-A architecture and exploring their potential applications is of great importance.

We have modified the introduction section of the manuscript on Page 4-5 to clarify this point. It now reads: “One type of D-A structured molecules, namely the D-A-D type, has attracted

considerable interest.⁴⁴⁻⁴⁶ By contrast, reports on A-D-A molecules as luminescent materials are relatively limited. In recent years, some reports suggest that A-D-A compounds can exhibit stronger charge transfer effects and light-harvesting capability compared to D-A-D structure, hence holding great potential in organic photovoltaics and biomedical applications.⁴⁷⁻⁵⁰”

Question 3: The authors used the term of “BioAIEgen” in the title. Please explain and further elaborate the advantages of such BioAIEgen in the manuscript, which is very important to help the readers to understand the superiority of the lignin-derived BioAIEgens.

Author reply: According to previous reports (*Angew. Chem. Int. Ed.*, **2020**, *59*, 9888-9907; *Nat. Commun.*, **2021**, *12*, 1773.), AIEgens or AIE-active nano agents obtained from natural resources including natural products or derivatives by modifying natural products can be termed as BioAIEgens. The lignin-derived platform chemicals are obtained from lignin, hence the AIEgens derived from them can be appropriately defined as BioAIEgens. Lignin is a renewable resource. However, its complex structure makes it difficult to controllably degrade, thereby limiting the application of lignin. On this basis, the effective and controllable conversion of lignin platform chemicals into value-added fine chemicals has become a research direction for high-value utilization of lignin (*ACS Cent. Sci.*, **2019**, *5*, 1707-1716). AIE materials have been widely used in value-added fields such as biomedical (*Chem. Sci.*, **2021**, *12*, 6488–6506) and optoelectronic devices (*Coord. Chem. Rev.*, **2022**, *473*, 214843) due to their unique optical properties, attracting significant attention. In this work, we rationally employ molecular design strategies to efficiently convert lignin platform chemicals into BioAIEgens using a concise and sustainable method, achieving regulation over their optical properties. The research method proposed in this work is expected to become an effective approach for the high-value utilization of lignin, while also providing a new resource for the development of AIE materials, ultimately achieving a ‘win-win’ situation.

As suggested, we have made modifications in the introduction section of the manuscript on page 4. The text now reads: “Bio-resource derived aggregation-induced emission luminogens (BioAIEgens, AIEgens or AIE-active nano agents obtained from natural resources including natural products or derivatives by modifying natural products)²⁷⁻³⁰ have raised considerable attention owing to their advantageous renewability, biodegradability, and biocompatibility, paving the way

to potential applications in biomedical,³⁴⁻³⁵ chemosensors/biosensors,^{36,37} and optoelectronic devices.^{38,39} As a natural biomass rich in aromatic and carbonyl subunits, lignin based building blocks display strong fluorescence and excellent self-assembly properties, which could make them serve as ideal precursors for the preparation of BioAIEgens.^{33,38-40} However, its complex structure makes it difficult to controllably degrade, thereby limiting the application of lignin. Current studies are usually limited to macromolecular BioAIEgens directly coming from lignin itself or structurally modified lignin based macromolecules, resulting in the fact that the luminescence mechanism and structure-property relationships are still unclear due to the complicated and uncontrollable macromolecular structures. Therefore, the efficient and controllable conversion of lignin platform chemicals into value-added fine chemicals has become a hot research topic for utilization of lignin. Therefore, obtaining BioAIEgens with well-defined structures through lignin depolymerization would provide an alternative strategy for the design of lignin-derived BioAIEgens.”

Question 4: Scheme 1 and Fig. 2a use R1 and R2 as variable substituents in their chemical structures, but not consistent. It is necessary to unify them for clear discussion.

Author reply: Thanks for the reviewer’s careful examination. We have made the corresponding modifications to Fig.6 in the manuscript, as shown below in Fig. R7.

Fig. R7 Molecular design of lignin-derived BioAIEgens

Question 5: Fig. 2 discusses the photophysical properties of **3d**, **3h**, and **3q** at molecular state, while Fig. 3 discusses the aggregate structures of **3p** and **3q** at solid state. However, the discussions are in one section without clear discrimination. Please clarify this point.

Author reply: In order to make the statements clearer and more logically structured, we have

divided the description into two paragraphs. We have modified the content of the second paragraph on Page 10-11 in the revised manuscript. It now reads: “In addition to the fluorescence behavior in solution and aggregate states, the photophysical properties in the solid state have been further studied. With the aim of investigating the structure-property relationships of solid-state fluorescence materials, single crystals of **3p** and **3q** have been obtained for further analysis. Similar to the solution and aggregate states, the unsubstituted solid state of **3d** still exhibits blue-violet fluorescence with a wavelength of 436 nm. **3p** and **3q** show a significant red-shift (**3p**: 493 nm; **3q**: 511 nm) compared to **3d**, probably due to the presence of electronically different substituents,^{64,65} thus enhancing the characteristics of CT effects.”

Question 6: Carbazole is commonly used to construct molecules with phosphorescent properties. Has the author explored the phosphorescence?

Author reply: As suggested, we have carried out the phosphorescence and lifetime tests of **3d**, **3p**, and **3q** in Fig. R8. The data show that they indeed exhibit phosphorescence with a lifetime range from 0.88 to 1.63 ms. Since the focus of our article is on the fluorescence properties, phosphorescence behaviors will be reported elsewhere at a later point and in more detail.

Fig. R8 (a-c) Phosphorescent spectra of **3d** (a), **3p** (b), and **3q** (c). (d-e) Phosphorescence decay curves of **3d** (d), **3p** (e), and **3q** (f).

Question 7: Please proof the references. The journal titles are not abbreviated for uniform, such

as Refs. 16 and 36.

Author reply: As suggested, we have rechecked and rectified the errors in the reference section of the manuscript.

REVIEWERS' COMMENTS

Reviewer #1 (Remarks to the Author):

The revised manuscript of Zhang and co-workers made great strides to address previous round of critique. Although synthetic novelty of the synthesis of carbazole derivatives are still a matter of debate in this manuscript, the authors tried to clarify the ambiguities with proper reasons which are reasonable and can shed light on their original submission. The novelty aspect has slightly increased and I am now open to accept the manuscript in Nature Communications, provided authors should meet minor addition on inclusion of the reaction outcome with N-H free indole in the substrate scope (Figure 2) and corresponding discussions in revised manuscript also.

'However, most synthetic strategies utilize expensive noble metal catalyst, excess oxidants and non-renewable substrates, which raises both atom economy and environmental concerns.' This comment though partially correct, however, there are considerable numbers of recent literature where related carbazoles prepared by 6π -electrocyclization, pinacol-type rearrangement, cycloaddition, electrophilic-cycloaromatization, etc. where simple Brønsted acid catalyst were utilized (Org. Lett. 2018, 20, 6920–6924, Org. Lett. 2016, 18, 23, 6200–6203, Org. Biomol. Chem., 2019, 17, 1822–1826, J. Org. Chem. 2020, 85, 13272–13279, J. Org. Chem. 2019, 84, 16003–16012, Chem. Eur. J. 2019, 25, 11521 – 11527, Org. Lett. 2017, 19, 22, 6140–6143, J. Org. Chem. 2017, 82, 6, 2935–2942; etc.). Synthesizing carbazoles while keeping N-H functional group unprotected is also a challenge as this group interfere with several other reactions leading to unwanted side product which authors also observed in this work, and nicely addressed in some of the above reports.

Thus in the manuscript, the line, 'However, most approaches suffer from the necessity to use expensive noble metal catalysts, the need of excess oxidants and the formation of non-renewable substrates, which raises atom economy and environmental concerns.' Should accompany with another line of discussion covering literature including organo-catalyzed methods of carbazole synthesis. As this line is one of the selling point of this manuscript from the synthesis point of view.

With these minor comments and addition, this reviewer suggests the acceptance of this manuscript for the publication.

Reviewer #2 (Remarks to the Author):

The authors have revised the manuscripts based on the peer-review comments of the Reviewers. From my side, the manuscript has been approved in many scientific aspects. That means it could be accepted for publication at this stage.

Reviewer #3 (Remarks to the Author):

The authors addressed all of my concerns. I recommended the publication now

Response to the comments of the reviewers

(NCOMMS-23-18138)

Towards BioAIEgens from lignin β -O-4 motifs as renewable resources

Tenglong Guo, Yuting Lin, Deng Pan, Xuedan Zhang, Wenqing Zhu, Xu-Min Cai, Genping Huang, Hua Wang, Dezhu Xu, Fritz E. Kühn, Bo Zhang, and Tao Zhang

Reviewer #1 (Remarks to the Author):

Q1: The revised manuscript of Zhang and co-workers made great strides to address previous round of critique. Although synthetic novelty of the synthesis of carbazole derivatives are still a matter of debate in this manuscript, the authors tried to clarify the ambiguities with proper reasons which are reasonable and can shed light on their original submission. The novelty aspect has slightly increased and I am now open to accept the manuscript in Nature Communications, provided authors should meet minor addition on inclusion of the reaction outcome with N-H free indole in the substrate scope (Figure 2) and corresponding discussions in revised manuscript also.

Author reply: As suggested, we have added this result in the manuscript on Page 7 and supporting information on Page 6. The text in the revised manuscript now reads: In contrast, the reaction of a N-H free indole derivative ((*E*)-3-(1*H*-indol-3-yl)-1-phenylprop-2-en-1-one) with **1d** was carried out, leading to ((*E*)-3-(1-(3-oxo-3-phenylpropyl)-1*H*-indol-3-yl)-1-phenylprop-2-en-1-one) as main product in 21% yield due to some side reactions (Supplementary Fig. 1). In order to obtain high selectivity of carbazole products, N-protected indole derivatives (**2a-2s**) are used as substrates. For example, a benzyl substituent on the N-R moiety in 3-alkenylated is tolerated, leading to 86% yield of **3e** (Fig. 2, entry 5).

Q2: 'However, most synthetic strategies utilize expensive noble metal catalyst, excess oxidants and non-renewable substrates, which raises both atom economy and

environmental concerns.’ This comment though partially correct, however, there are considerable numbers of recent literature where related carbazoles prepared by 6π -electrocyclization, pinacol-type rearrangement, cycloaddition, electrophilic-cycloaromatization, etc. where simple Brønsted acid catalyst were utilized (Org. Lett. 2018, 20, 6920–6924, Org. Lett. 2016, 18, 23, 6200–6203, Org. Biomol. Chem., 2019, 17, 1822–1826, J. Org. Chem. 2020, 85, 13272–13279, J. Org. Chem. 2019, 84, 16003–16012, Chem. Eur. J. 2019, 25, 11521 – 11527, Org. Lett. 2017, 19, 22, 6140–6143, J. Org. Chem. 2017, 82, 6, 2935–2942; etc.). Synthesizing carbazoles while keeping N-H functional group unprotected is also a challenge as this group interfere with several other reactions leading to unwanted side product which authors also observed in this work, and nicely addressed in some of the above reports.

Thus in the manuscript, the line, ‘However, most approaches suffer from the necessity to use expensive noble metal catalysts, the need of excess oxidants and the formation of non-renewable substrates, which raises atom economy and environmental concerns.’ Should accompany with another line of discussion covering literature including organo-catalyzed methods of carbazole synthesis. As this line is one of the selling point of this manuscript from the synthesis point of view.

Author reply: As suggested, we have added some discussion into the revised manuscript on Page 5 and improved the corresponding discussion. The text in the revised manuscript now reads: Moreover, extensive synthetic strategies of the functionalized carbazoles focus on intermolecular cross-coupling reactions between C-H/C-X bonds (X= halo, N, O, C, etc),⁵⁶ oxidative intramolecular C-H/C-H cross coupling of prefunctionalized diarylamines,⁵⁷ the construction of a benzene ring upon substituted indoles through transition metal catalysis⁵⁸ or Brønsted acid catalysis.⁵⁹⁻⁶¹ However, most approaches suffer from the need of excess oxidants, multi-step reactions and the formation of non-renewable substrates, which raises atom economy and environmental concerns.

Reviewer #2 (Remarks to the Author):

The authors have revised the manuscripts based on the peer-review comments of the Reviewers. From my side, the manuscript has been approved in many scientific aspects.

That means it could be accepted for publication at this stage.

Reviewer #3 (Remarks to the Author):

The authors addressed all of my concerns. I recommended the publication now